

# Early-branching euteleost relationships: areas of congruence between concatenation and coalescent model inferences

Matthew A. Campbell[1,2], Michael E. Alfaro[3], Max Belasco[3] and J. Andrés López[4,5]

[1] Department of Biology and Wildlife, University of Alaska Fairbanks, Fairbanks, AK, United States of America

[2] Department of Ecology and Evolutionary Biology, University of California, Santa Cruz, CA, United States of America

[3] Department of Ecology and Evolutionary Biology, University of California Los Angeles, Los Angeles, CA, United States of America

[4] School of Fisheries and Ocean Sciencs, University of Alaska Fairbanks, Fairbanks, AK, United States of America

[5] University of Alaska Museum, Fairbanks, AK, United States of America

Corresponding author
Matthew A. Campbell,
drmaccampbell@gmail.com

## ABSTRACT

Phylogenetic inference based on evidence from DNA sequences has led to significant strides in the development of a stable and robustly supported framework for the vertebrate tree of life. To date, the bulk of those advances have relied on sequence data from a small number of genome regions that have proven unable to produce satisfactory answers to consistently recalcitrant phylogenetic questions. Here, we re-examine phylogenetic relationships among early-branching euteleostean fish lineages classically grouped in the Protacanthopterygii using DNA sequence data surrounding ultraconserved elements. We report and examine a dataset of thirty-four OTUs with 17,957 aligned characters from fifty-three nuclear loci. Phylogenetic analysis is conducted in concatenated, joint gene trees and species tree estimation and summary coalescent frameworks. All analytical frameworks yield supporting evidence for existing hypotheses of relationship for the placement of *Lepidogalaxias salamandroides*, monophyly of the Stomiatii and the presence of an esociform + salmonid clade. *Lepidogalaxias salamandroides* and the Esociformes + Salmoniformes are successive sister lineages to all other euteleosts in the majority of analyses. The concatenated and joint gene trees and species tree analysis types produce high support values for this arrangement. However, inter-relationships of Argentiniformes, Stomiatii and Neoteleostei remain uncertain as they varied by analysis type while receiving strong and contradictory indices of support. Topological differences between analysis types are also apparent within the otomorph and the percomorph taxa in the data set. Our results identify concordant areas with strong support for relationships within and between early-branching euteleost lineages but they also reveal limitations in the ability of larger datasets to conclusively resolve other aspects of that phylogeny.

## INTRODUCTION

Phylogenomic datasets comprising hundreds to thousands of genome segments produced through high throughput sequencing technology have shown promise to resolve difficult phylogenetic problems (e.g., *Faircloth et al., 2013*; *Faircloth et al., 2012*; *Gilbert et al., 2015*; *Harrington et al., 2016*; *Lemmon & Lemmon, 2013*). At the same time, novel and refined inference tools including implementations of the multispecies coalescent model to address incomplete lineage sorting (ILS) through Gene Trees-to-Species Tree (GT-ST) methods (*Knowles & Kubatko, 2011*) continue to extend the power and complexity of phylogenetic research. Despite these advances in genomic-scale dataset production and phylogenetic inference, difficult areas of the tree of life remain unresolved (*Delsuc, Brinkmann & Philippe, 2005*; *Pyron, 2015*; *Rokas & Carroll, 2006*). Relationships among early-branching euteleost lineages remain nebulous (e.g., *Betancur-R et al., 2013*; *Campbell et al., 2013*; *Li et al., 2010*; *Near et al., 2012*) and stand out as one of the most contentious regions of the fish tree of life. Although this question has been studied from morphological and molecular perspectives consensus has yet to emerge.

The name Euteleostei was first applied to a diverse group of fishes that includes all teleosts outside of the superorders Elopomorpha, Osteoglossomorpha and Clupeomorpha by phyletic analysis (*Greenwood et al., 1967*; *Greenwood et al., 1966*). *Rosen (1985)* excluded esocoids from the Euteleostei based on cladistic analyses of morphological characters, while *Johnson & Patterson (1996)* included esocoids but excluded ostariophysans. Subsequent phylogenetic studies of mitochondrial (e.g., *López, Chen & Ortí, 2004*; *Lavoué et al., 2008*) and nuclear DNA (e.g., *Betancur-R et al., 2013*; *Near et al., 2012*) supported a monophyletic Euteleostei including esocoids but excluding Ostariophysi and the Alepocephaliformes (previously classified in Argentiniformes nested in the Euteleostei).

Recent phylogenetic studies based on molecular evidence consistently support the monophyly of five major euteleost lineages (*Betancur-R et al., 2013*; *Campbell et al., 2013*; *Li et al., 2010*; *Near et al., 2012*): (1) a clade formed by Esociformes and Salmoniformes; (2) the Stomiatii *sensu Betancur-R et al. (2013)* consisting of Osmeriformes (excluding Galaxiiformes) and Stomiiformes; (3) the Argentiniformes (excluding the Alepocephaliformes); (4) the Galaxiiformes (excluding *Lepidogalaxias*); and (5) the Neoteleostei. In addition, these studies agree on placing the monotypic *Lepidogalaxias* as the sister group of all other euteleosts. Aside from the placement of *Lepidogalaxias,* there is little congruence among different studies regarding relationships among the five lineages (e.g., *Betancur-R et al., 2013*; *Campbell et al., 2013*; *Li et al., 2010*; *Near et al., 2012*). The early branching patterns of euteleosts are still in need of further study and represent a difficult problem for traditional morphological and molecular phylogenetics.

Here we apply the "new and general theory of molecular systematics" (*Edwards, 2009*) to examine early-branching euteleost relationships using multi-locus datasets generated by targeted enrichment of conserved nuclear DNA sequences. Concatenated and GT-ST phylogenetic inference frameworks are used to assess the stability and strength of evidence for alternative arrangements in this poorly resolved section of the fish tree of life.

## MATERIAL AND METHODS

### Taxon and character sampling

We targeted species representing five of the six major euteleost lineages as well as several non-euteleost outgroups (Table S1). We prepared genomic DNA libraries with 500–600 bp inserts by shearing total genomic DNA extracts to size using a sonicator (Diagenode, Inc) and ligating a set of custom-indexed Illumina Tru-Seq compatible adapters (*Faircloth & Glenn, 2012*) to the sheared DNA using reagents from a library preparation kit (KapaBiosystems, Inc.). Adapter-ligated DNA was amplified with 16–18 cycles of PCR. To obtain sequences from homologous loci across the taxonomic sample, we performed targeted enrichment of ultraconserved element (UCEs) loci shared among acanthopterygians following protocols outlined in *Faircloth et al. (2013)*. We modified the capture protocol by pooling eight, indexed sequencing libraries at equimolar ratios prior to enrichment and performing 12–16 cycles of PCR-recovery after enrichment. Following the enrichment procedure, we quantified enriched, amplified libraries using a commercial qPCR quantification kit (KapaBiosystems, Inc.), and we prepared an equimolar pool of pooled libraries for sequencing on an Illumina HiSeq 2500 instrument using 100 base pair, paired-end sequencing chemistry in rapid run mode (UCLA Neuroscience Genomics Core). To extend our taxon sampling, we included previously published UCE data (*Faircloth et al., 2013*) in our analyses (Table S1).

### Raw sequence data processing

Demultiplexed reads were edited for length, overall quality and adapter contamination using Trimmomatic v. 0.32 (*Bolger, Lohse & Usadel, 2014*). We assembled a subset of cleaned reads across various kmers with Velvet v. 1.2.10 (*Zerbino & Birney, 2008*) to establish a range of suitable kmers for assembly. We then assembled sequences for each species using two different approaches. For non-salmonids, we assembled reads using VelvetOptimiser v. 2.2.5 across the optimal range of kmers we identified (57 to 83). For salmonids, assemblies from Velvet were produced for each value between 57 and 83. However, as the optimization performed by VelvetOptimiser is designed for haploid or diploid organisms, an alternative selection criterion of the maximum number of single copy UCE loci was chosen to accommodate the effect of ancestral polyploidy in salmonid genomes (*Allendorf & Thorgaard, 1984*). A single dataset assembly was retained downstream analyses from each alternative approach to data assembly. We identified homologous UCE loci and prepared sequences for alignment with the PHYLUCE pipeline (*Faircloth, 2016*). During orthology assessment, the PHYLUCE package screens for and removes from analysis reciprocally duplicate enriched loci, which may represent paralogs.

### Alignment and phylogenetic analysis

Following orthology assessment, the taxon set consisted of thirty-four Operational Taxonomic Units (OTUs) representing outgroups and basal euteleost lineages. We ensured this taxon set included loci sequenced in at least 31 of the 34 OTUs. We aligned data from all loci in with MAFFT v. 7.130b (*Katoh et al., 2002*) through the PHYLUCE pipeline (*Faircloth, 2016*).

We analyzed the 34-OTU dataset under the Maximum-Likelihood (ML) framework as implemented in RAxML v. 8.1.24 (*Stamatakis, 2014*). Each UCE locus was modeled as a partition evolving under the general time reversible (GTR) model of sequence evolution with gamma distributed rate variation ($\Gamma$). We set ML pseudoreplicate searches to automatically stop when stable bootstrap indices were detected (autoMRE). A joint gene trees and species tree estimation was conducted in a Bayesian framework with *BEAST (*Heled & Drummond, 2010*) as implemented in BEAST v. 2.1.3 (*Drummond et al., 2012*). We analyzed data using a constant coalescent model under a Hasegawa-Kishino-Yano (HKY) model of sequence evolution with a four-category gamma distributed rate variation ($\Gamma$) and empirical base frequencies to each locus. Convergence and sufficient effective sample sizes (ESSs, >200) of all parameters were reached by combining three chains of 800 million generations with 40% burn-in. Two additional analyses were conducted. To verify that partitioning in the ML analysis by gene does not influence early-branching euteleost relationships and support values, objective partitioning was investigated. To verify that the use of the coalescent model in *BEAST resulted in an alternative arrangement of early-branching euteleost lineages, not the choice of nucleotide evolution model, a concatenated Bayesian analysis with BEAST 2 with the same nucleotide evolution model for each UCE locus as *BEAST (HKY$+\Gamma$ with empirical base frequencies) was undertaken (Document S1).

## Topology tests and occurrence of particular arrangements in the Bayesian tree posterior sample

To determine the significance of UCE evidence corroborating or refuting alternative phylogenetic arrangements, we tested the following topologies resulting from concatenated and GT-ST analysis against each other: (1) the best-scoring ML topology; (2) the consensus species-tree topology from *BEAST; and (3) a Protacanthopterygii *sensu Betancur-R et al. (2013)* as the sister lineage to the Stomiatii. A best scoring ML tree (1 from above) and constrained trees (2 and 3 from above) were generated with RAxML v. 8.2.3 partitioned by UCE using a GTR$+\Gamma$ model of nucleotide evolution. We tested the trees against each other by generating per site likelihoods with RAxML and analyzing the output with CONSEL v. 0.20 (*Shimodaira & Hasegawa, 2001*). CONSEL implements several hypothesis tests allowing a more rigorous comparison between alternative hypotheses than solely comparing likelihood values.

As the *BEAST posterior tree presented as the consensus species-tree topology represents the combination of many different species trees, we searched the combined post burn-in posterior tree sample from the separate *BEAST chains (180,003 trees) for alternative phylogenetic hypotheses to determine if the *BEAST algorithm considered these alternatives. The *BEAST posterior tree sample was searched for the best scoring ML topology and a monophyletic Protacanthopterygii *sensu Betancur-R et al. (2013)* with Python scripts (*Moravec, 2015*).

## Summary coalescent analyses

To further examine the potential impacts of small regions in concatenated alignments on key relationships differing between RAxML concatenated analyses and the *BEAST

species tree, we examined our data through summary coalescent analyses. These methods function on independently estimated gene trees, therefore decreasing the influence of small regions in concatenated analyses (*Shen, Hittinger & Rokas, 2017*) and accounting for ILS. Furthermore, *BEAST and BEAST assume a molecular clock that may contribute to discrepancies between these analyses and RAxML analyses. For each of the UCE loci, we generated a gene tree with nucleotide evolution modeled with the GTR nucleotide evolution model and gamma distributed rate variation (Γ) with RAxML version 8.0.19. A set of rooted gene trees were also generated by specifying *Polypterus senegalus* as the outgroup. Four summary coalescent analysis frameworks were applied to the gene trees. The Accurate Species TRee ALgorithm (ASTRAL) and Neighbor Joining species tree (NJ*st*) tree methods accept unrooted trees with missing taxa. The unrooted gene trees were analyzed program ASTRAL version 4.10.12 (*Mirarab et al., 2014*; *Mirarab & Warnow, 2015*) and with NJ*st* (*Liu & Yu, 2011*) via the Species TRee Analysis Web server (STRAW) (*Shaw et al., 2013*). The rooted trees were analyzed with both STAR (*Liu et al., 2009*) and MP-EST (*Liu, Yu & Edwards, 2010*) through STRAW.

## RESULTS

### Characteristics of UCE dataset

Following orthology assessment and filtering for loci not present in 31 of 34 OTUs, the dataset is composed of a total of 53 UCE loci, 17,957 characters, 9,576 distinct alignment patterns and 22.11% gaps or missing data. We present details of the number of UCE loci recovered for each taxon, the average length of UCE matching contigs, average coverage of contigs matching UCEs and number of duplicate loci removed in Table S1. The assemblies and alignment are available within the Data S1.

### Early-branching euteleost relationships

Concatenated ML analysis supports a monophyletic Euteleostei, excluding Ostariophysi and Alepocephaliformes (Bootstrap Support [bs] = 100%). Figure 1 shows the inferred branching pattern among main euteleost groups from the 34-OTU dataset. Relationships among main euteleost lineages in the concatenated ML topology are (*Lepidogalaxias*, ((Esociformes, Salmoniformes), (Argentiniformes, (Stomiatii, Neoteleostei)))) with all nodes among those lineages receiving strong support (bs = 100%).

  GT-ST analysis of the dataset in *BEAST indicates a monophyletic Euteleostei with high support, posterior probability (*pp*) = 1.00 (Fig. 2). A topology of (*Lepidogalaxias*, ((Esociformes, Salmoniformes), ((Argentiniformes, Stomiatii), Neoteleostei)))) is generated in this analysis. Support values for the placement of main euteleost lineages are high throughout the consensus tree. The placement of *Lepidogalaxias* and the Esociformes + Salmoniformes receive very high support (*pp* = 1.00). Argentiniformes + Stomiatii as the sister lineage of the neoteolosts received strong support (*pp* = 0.99). A sister relationship between the Argentiniformes and Stomiatii was also well supported ( *pp* = 0.96). The GT-ST and ML inferred phylogenies differ on the relationships among argentiniforms, stomiatians and neoteleosts.

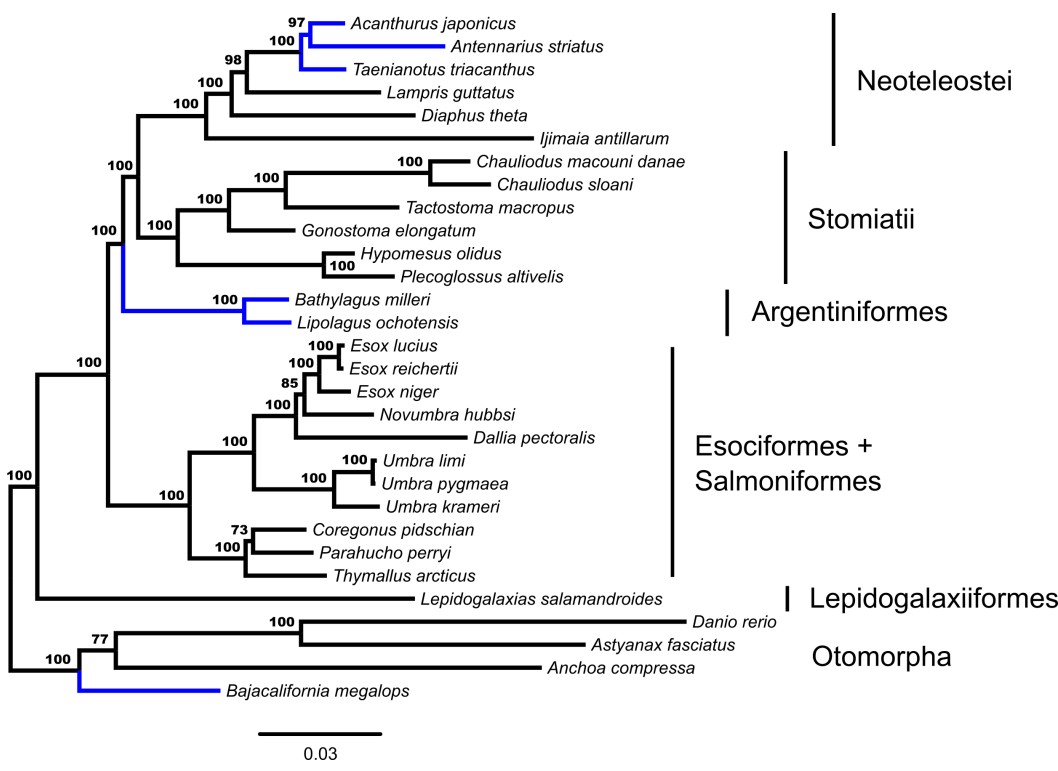

**Figure 1** **Phylogenetic tree from fifty-three ultraconserved element (UCE) loci generated in a concatenated framework with RAxML.** Each locus is designated as a partition and modeled under a GTR + Γ model of nucleotide evolution. Values from automatic stopping of bootstrap replicates are indicated at each node. The tree is rooted by *Polypterus senegalus*. *Polypterus senegalus*, *Amia calva*, *Osteoglossum bicirrhosum* and *Pantodon buchholzi* are omitted from figure. Early-branching euteleost taxa are labeled and indicated. From the Neoteleostei, Ateleopodiformes and Acanthuriformes drawings are included. Placements of taxa that are different from the GT-ST topology (Fig. 2) are indicated in blue.

Through the additional concatenated analyses presented in the Document S1, conflicts between ML and GT-ST results presented in Figs. 1 and 2 are shown to be the product of the distinct analytical frameworks and do not result from how data were modeled. The additional concatenated analyses in Document S1 show identical branching patterns for main early-branching euteleost lineages to the concatenated ML analysis presented in Fig. 1 with high support values. Retaining the same model but changing the partitioning strategy with RAxML demonstrates that the inferred phylogeny from the ML analysis presented in Fig. 1 is not sensitive to partitioning (Fig. S1). Not implementing a *BEAST model, while retaining the same nucleotide evolution and partitioning scheme for a concatenated analysis with BEAST 2 also produces a phylogeny (Fig. S2) with the branching of main early-branching euteleost lineages matching that of the concatenated ML analysis presented in Fig. 1, not the *BEAST GT-ST analysis presented in Fig. 2. Consequently, the topological differences between phylogenies shown in Fig. S2 and Fig. 2 may be attributed to whether a concatenated or coalescent approach is implemented. There are three key locations in the inferred trees that differ: (1) within the Otomorpha, the placement of

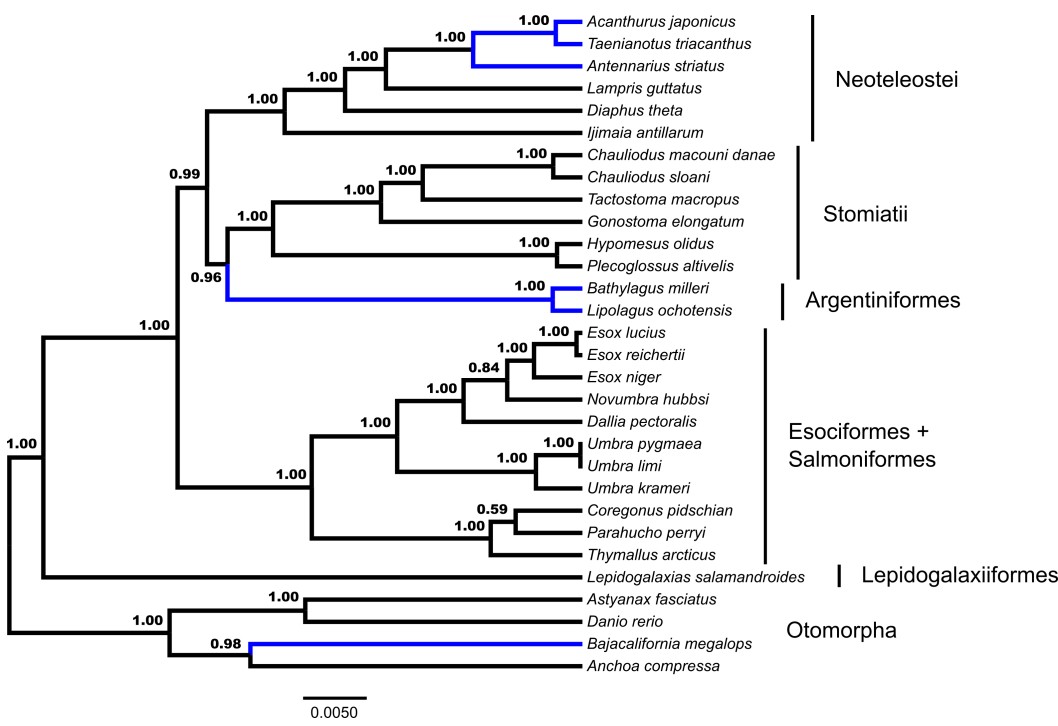

**Figure 2 Species tree from *BEAST.** Fifty-three ultraconserved element (UCE) loci are modeled under an HKY model of nucleotide sequence evolution with a four category gamma distribution characterizing rate variation among sites ($\Gamma$). Each model of sequence evolution has independent model parameters. This tree represents the combination of three independent *BEAST runs with the posterior probability of each node indicated. Early-branching euteleost lineages are labeled and indicated. Images of neoteleost lineages from Acanthuriformes and Ateleopodiformes are also included. The tree is rooted by *Polypterus senegalus*. *Polypterus senegalus*, *Amia calva*, *Osteoglossum bicirrhosum* and *Pantodon buchholzi* are omitted from figure. Placements of taxa that are different from the concatenated topology (Fig. 1) are indicated in blue.

*Bajacalifornia megalops*, (2) the arrangement of early-branching euteleost lineages, and (3) the arrangement of the three percomorph lineages.

## Topology tests and occurrence of particular arrangements in the Bayesian tree posterior sample

Testing with CONSEL indicates the best-scoring ML tree, with a topology of (*Lepidogalaxias*, ((Esociformes, Salmoniformes), (Argentiniformes, (Stomiatii, Neoteleostei)))), is significantly better than the topology generated by GT-ST analysis with both the approximately unbiased test ($p = 1 \times 10^{-5}$) and the weighted Shimodaira-Hasegawa test ($p = 1 \times 10^{-3}$). A monophyletic assemblage of protacanthopterygian taxa *sensu Betancur-R et al. (2013)* sister to the Stomiatii is significantly worse than the best-scoring ML tree with both the approximately unbiased test ($p = 8 \times 10^{-6}$) and the weighted Shimodaira-Hasegawa test ($p = 1 \times 10^{-4}$). The posterior set of 180,003 trees generated by *BEAST did not include a single occurrence of either the ML best tree topology or a monophyletic Protacanthopterygii *sensu Betancur-R et al. (2013)*.

**Table 1** Comparison of results from all phylogenetic analyses presented in this study of topological differences within the Otomorpha, the early-branching euteleost lineages and the Percomorpha. Concatenation or Gene Trees-to-Species Tree (GT-ST) framework and analysis program are indicated in the Method column. If a tree is presented, it is indicated in the Figure column. The three key topological differences determined between concatenation in RAxML presented in Fig. 1 and *BEAST presented in Fig. 2 are indicated and color-coded to analysis method. Results consistent with the RAxML results are blue and those consistent with the *BEAST analysis are orange. A unique topology is not color-coded for early-branching euteleosts present only in the results from MP-EST.

| Method | Figure | Otomorpha | Early-Branching Euteleosts | Percomorpha |
|---|---|---|---|---|
| Concatenation RAxML | Fig. 1 | (Bajacalifornia, (Anchoa, (Astyanax, Danio))) | (Lepidogalaxias, (Esociformes + Salmoniformes, (Argentiniformes, (Stomiatii, Neoteleosteii)))) | (Taenianotus, (Acanthurus, Antennarius)) |
| GT-ST *BEAST | Fig. 2 | ((Anchoa, Bajacalifornia), (Astyanax, Danio)) | (Lepidogalaxias, (Esociformes + Salmoniformes, ((Argentiniformes, Stomiatii), (Neoteleosteii)))) | (Antennarius, (Acanthurus, Taenianotus)) |
| Concatenation RAxML | Fig. S1 | (Bajacalifornia, (Anchoa, (Astyanax, Danio))) | (Lepidogalaxias, (Esociformes + Salmoniformes, (Argentiniformes, (Stomiatii, Neoteleosteii)))) | (Taenianotus, (Acanthurus, Antennarius)) |
| Concatenation BEAST | Fig. S2 | ((Anchoa, Bajacalifornia), (Astyanax, Danio)) | (Lepidogalaxias, (Esociformes + Salmoniformes, (Argentiniformes, (Stomiatii, Neoteleosteii)))) | (Antennarius, (Acanthurus, Taenianotus)) |
| GT-ST ASTRAL | Fig. 3 | (Bajacalifornia, (Anchoa, (Astyanax, Danio))) | (Lepidogalaxias, (Esociformes + Salmoniformes, (Argentiniformes, (Stomiatii, Neoteleosteii)))) | (Antennarius, (Acanthurus, Taenianotus)) |
| GT-ST NJst | | (Bajacalifornia, (Anchoa, (Astyanax, Danio))) | (Lepidogalaxias, (Esociformes + Salmoniformes, (Argentiniformes, (Stomiatii, Neoteleosteii)))) | (Antennarius, (Acanthurus, Taenianotus)) |
| GT-ST STAR | | ((Anchoa, Bajacalifornia), (Astyanax, Danio)) | (Lepidogalaxias, (Esociformes + Salmoniformes, (Argentiniformes, (Stomiatii, Neoteleosteii)))) | (Antennarius, (Acanthurus, Taenianotus)) |
| GT-ST MP-EST | | (Bajacalifornia, (Anchoa, (Astyanax, Danio))) | (Lepidogalaxias, (Argentiniformes, (Esociformes + Salmoniformes, (Stomiatii, Neoteleostei)))) | (Antennarius, (Acanthurus, Taenianotus)) |

## Summary coalescent analyses

Fifty-three unrooted gene trees were analyzed by ASTRAL and NJst, the requirement of rooting by *Polypterus senegalus* reduced the number of available gene trees that are rooted to forty-four for STAR and MP-EST. The summary coalescent results vary with regards to the three key differences identified previously as presented in Figs. 1 and 2 (Table 1). ASTRAL (Fig. 3) and NJst resolves the otomorphs as found by RAxML concatenation analyses, the early-branching euteleost relationships as in RAxML and BEAST concatenation, while the percomorph taxa are found in an arrangement from *BEAST and BEAST. ASTRAL posterior probability values for the placement of the Argentiniformes and Stomiatii + Neoteleosteii are low (0.58 and 0.56 respectively). STAR resolves the otomorphs and percomorphs as BEAST and *BEAST analyses, but early-branching euteleosts match the concatenated RAxML results. MP-EST matches the concatenated RAxML results for otomorphs and BEAST / *BEAST for the percomorphs while producing a unique topology of early-branching euteleost lineages.

## DISCUSSION

### Hypotheses of early-branching euteleost relationships

Our phylogenomic analysis provides strong support for relationships of early diverging euteleosts that consist of *Lepidogalaxias* and esociforms + salmoniforms as successive sister lineages to a clade containing argentiniforms, stomiatiids and neoteleosts. Despite the most intensive character sampling of this group to date, our analyses do not resolve two conflicting hypotheses for relationships among the Argentiniformes, Stomiatii

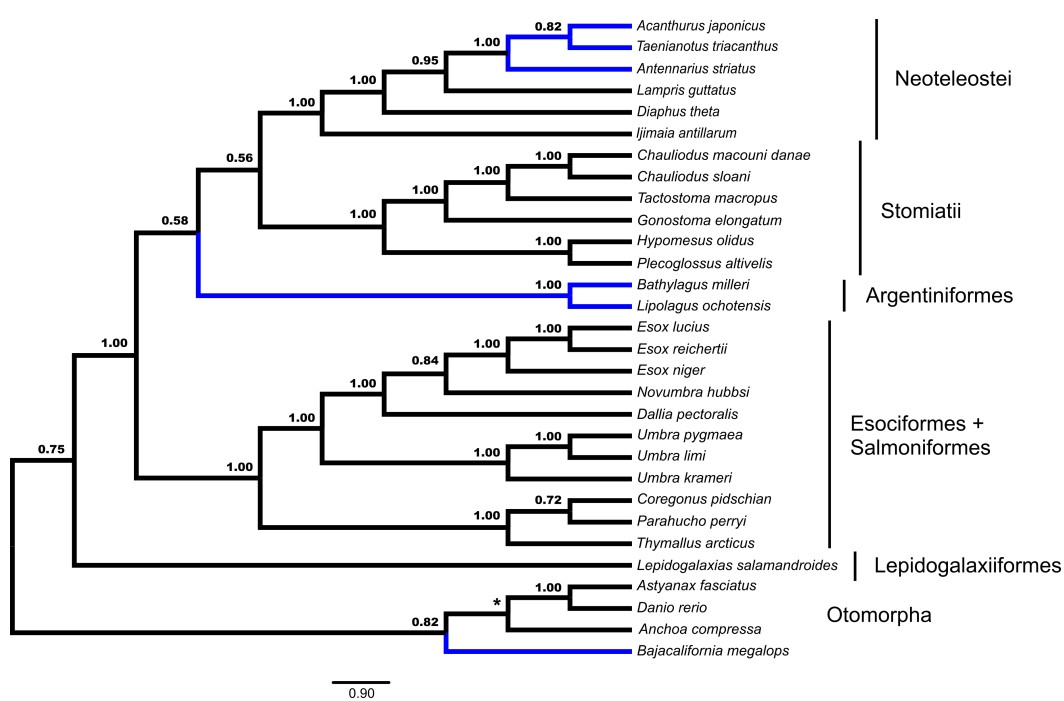

**Figure 3** **Species tree from ASTRAL.** Gene trees from fifty-three ultraconserved element (UCE) loci were separately generated with a GTR + Γ model of nucleotide evolution with RAxML. All fifty-three gene trees were then processed by the ASTRAL algorithm which rooted each tree arbitrarily with *Anchoa compressa*. Posterior probability for each node is indicated for all nodes except for the node involved with arbitrary rooting by ASTRAL for which no posterior probability was provided and is indicated with an asterisk (*). Early-branching euteleost lineages are labeled and indicated. Images of neoteleost lineages from Acanthuriformes and Ateleopodiformes are also included. The species tree from ASTRAL was re-rooted by *Polypterus senegalus*. *Polypterus senegalus*, *Amia calva*, *Osteoglossum bicirrhosum* and *Pantodon buchholzi* are omitted from figure. Placement of lineages that were shown to be different in Figs. 1 and 2 are indicated in blue.

and Neoteleostei. The concatenated ML derived topology resolves argentiniforms and stomatiids as successive sister lineages to the neoteleosts as do most summary coalescent methods, while the *BEAST GT-ST analysis recovers an argentiniform + stomatiids clade as the sister group to neoteleosts.

Combined, our analyses yield strong support for the Esociformes + Salmoniformes clade, which has found robust and consistent support in molecular phylogenetic studies (*López, Chen & Ortí, 2004*), reviewed by *Campbell et al. (2013)*, despite weak or conflicting evidence from morphology (*Johnson & Patterson, 1996*; *Wilson & Williams, 2010*). We also recover the Stomiatii (Osmeriformes + Stomiiformes) with high support values in both analyses in this study. On the other hand, we do not find a close relationship between the clade of Esociformes + Salmoniformes and any other major group of early-branching euteleosts such as Argentiniformes (*Near et al., 2012*). Instead, as shown in mitogenomic phylogenies (*Campbell et al., 2013*; *Inoue et al., 2003*) or analyses of combined mitochondrial and nuclear data (*Burridge et al., 2012*), we find Esociformes and Salmoniformes as sister to all other euteleosts in the study, with the exclusion of *Lepidogalaxias*.

## Support for hypotheses of early-branching euteleost lineages

Unlike other molecular (and morphological) studies of the euteleost phylogeny (e.g., *Betancur-R et al., 2013*; *Li et al., 2010*; *Near et al., 2012*), our conflicting topologies are strongly supported by both bootstrap values and Bayesian posterior probabilities.

Earlier studies typically yield low or moderate support for relationships along this section of the teleost phylogeny backbone. For example, the placement of the Argentiniformes and Salmoniformes + Esociformes sister to the remaining three major euteleost lineages (Stomiati, Galaxiiformes, and neoteleosts) receives a bootstrap support value between 70–89% in *Near et al. (2012)*. Other nodes supporting the branching order of the five major euteleost lineages are supported by 90–99% bootstrap support values. The bootstrap support for a sister Protacanthopterygii *sensu Betancur-R et al. (2013)* and Stomiatii is 36%, and the monophyly of Stomiatii receives a bootstrap support of 73% in the hypothesis presented by *Betancur-R et al. (2013)*. In a mitochondrial genome based study, a sister relationship of Argentiniformes to the Salmoniformes + Esociformes receives a bootstrap support of 74% (*Li et al., 2010*). In the same study, the Argentiniformes, Salmoniformes and Esociformes are the sister lineage of the Stomiatii, supported by an 81% bootstrap support value (*Li et al., 2010*).

While we find uncharacteristically high support for branching relationships among all of the four major euteleost lineages represented in this study in a concatenated ML framework, gauging the significance of high bootstrap values in analyses of large data matrices is problematic. Bootstrap values may be high even with conflict or systematic error (*Felsenstein, 1978*; *Hillis & Bull, 1993*; *Huelsenbeck, 1997*). Concatenated ML phylogenomic analysis has previously been demonstrated with 1,070 genes in yeasts to produce 100% bootstrap support for all internodes, despite incorrect branching likely present (*Salichos & Rokas, 2013*). The *BEAST GT-ST analysis also produces high support values; however, posterior probability values themselves are both conditioned on the model of evolution and are not guaranteed to have good frequentist statistical behavior (*Alfaro, Zoller & Lutzoni, 2003*; *Alfaro & Holder, 2006*) and may be misleading under certain conditions (*Suzuki, Glazko & Nei, 2002*; *Salichos & Rokas, 2013*). Another potential issue in phylogenetics is that a few loci may substantially influence results in concatenated analyses (*Shen, Hittinger & Rokas, 2017*). To mitigate the influence of small regions of large effect in a concatenated analysis, we evaluated individual gene trees with four summary coalescent analyses. Three of the four summary coalescent analyses were in agreement with the RAxML concatenated analyses and the concatenated BEAST analysis hypothesis of early-branching euteleost lineages. The MP-EST analysis produced a unique result regarding early-branching euteleost lineages among all analyses. While the otomorph taxa are not consistently found to have a certain topology in the summary coalescent methods, three of the four analyses match the concatenated RAxML results. The summary coalescent methods indicate that the early-branching euteleost topology of *BEAST may be a result of the underlying assumption of a molecular clock in this analysis or other issue. The consistent results for percomorph relationships across GT-ST methods suggests that a particular locus or subset of loci and/or incomplete lineage sorting is affecting the concatenated analysis in RAxML.

## Hypothesis testing and alternative topologies in the Bayesian posterior tree sample

In a hypothesis testing framework, the optimal topology from the *BEAST GT-ST framework is a significantly worse fit compared to the concatenated ML best tree. Conversely the concatenated ML best tree topology is absent from the 180,003 posterior trees produced in the *BEAST GT-ST analysis. Combined, these demonstrate that strong conflicting signal underlies these topological differences. Recent studies have alternatively suggested that concatenation may perform better than GT-ST when individual loci are not long enough to resolve phylogenies (*Gatesy & Springer, 2014*), that concatenation and GT-ST methods should behave similarly under a range of conditions (*Tonini et al., 2015*), and that phylogenomic scale data sets may exacerbate problems of model misspecification (*Liu et al., 2015*). For additional discussion around these issues see also *Edwards et al. (2016)* and *Springer & Gatesy (2016)*. At present, the relationships of the argentiniforms and stomatiids to neoteleosts remain unclear and may depend strongly on the inclusion of the Galaxiidae. The placement of galaxiids has been unstable (*Betancur-R et al., 2013*; *Burridge et al., 2012*; *Campbell et al., 2013*; *Ishiguro, Miya & Nishida, 2003*; *Li et al., 2010*; *López, Chen & Ortí, 2004*; *Near et al., 2012*), although independent studies (e.g., *Campbell et al., 2013*; *Near et al., 2012*) suggest that galaxiids may be the sister lineage of the Neoteleostei.

## Lack of evidence for the monophyly of protacanthopterygians

The Protacanthopterygii is an historically important taxon of early-branching euteleosts with its definition and content repeatedly modified (e.g., *Greenwood et al., 1966*; *Johnson & Patterson, 1996*; *Lauder & Liem, 1983*; *Rosen, 1973*; *Rosen & Greenwood, 1970*; *Rosen & Patterson, 1969*). Protacanthopterygian monophyly as defined by morphology (e.g., *Johnson & Patterson, 1996*) was questioned by molecular phylogenetics (*Ishiguro, Miya & Nishida, 2003*). More recently, the Protacanthopterygii was redefined by *Betancur-R et al. (2013)* with molecular phylogenetics (bs of 37%) containing the Argentiniformes, Galaxiiformes, Esociformes and Salmoniformes. Although we were unable to obtain representatives of Galaxiiformes, our analyses demonstrate that the Argentiniformes are not most closely related to the Esociformes + Salmoniformes. A topology test using available taxa in this dataset further rejected the Protacanthopterygii *sensu Betancur-R et al. (2013)*.

## CONCLUSIONS

The two approaches (concatenation and GT-ST) implemented in this study indicate large areas of congruence in topology resolving several relationships within early-branching euteleost relationships. However, the disagreements highlight some of the potential caveats in resolving all relationships of the early-branching euteleosts. We report the first study using a joint GT-ST method to examine the question of early-branching euteleost relationships. A joint estimation of species tree and gene trees can be preferred over summary methods (*Gatesy & Springer, 2014*) and produced a slightly different hypothesis of relationships when compared to concatenated analyses. A test of topology rejects the species-tree topology over the best scoring concatenated ML topology. Likewise, posterior support for the *BEAST Bayesian species tree hypothesis is high for early-branching

euteleost nodes, indicating very few occurrences of alternative topologies in the tree search. For major euteleost lineages, relationships among Argentiniformes, Neoteleostei and Stomiatii differed in the results of concatenated ML and Bayesian joint GT-ST analyses. This is in line with previous research on early-branching euteleost relationships. The lack of agreement between studies of early-branching euteleost relationships may be caused by short internode distances deep in the evolutionary past, leading to the formation and preservation of few informative characters linking these old lineages. A related but less likely possibility is that short internodes associated with very rapid diversification created conditions conducive to pervasive ILS at the base of the euteleost radiation resulting in conflicting histories across euteleost genomes and incongruent results between studies of early-branching euteleost relationships.

We evaluated identical datasets under concatenated and GT-ST frameworks and found three areas of incongruence: (1) argentiniform sister lineage, (2) the placement of the alepocephaliform lineage *Bajacalifornia*, and (3) the arrangement of the three neoteleost lineages *Antennarius*, *Acanthurus* and *Taenianotus*. The percomorph taxa (*Antennarius*, *Acanthurus* and *Taenianotus*) belong in a set of fish lineages whose relationship have been particularly difficult to elucidate (*Nelson, 1989*). The incongruent inferences we observed between approaches may be differential effects of ILS on coalescent versus non-coalescent phylogenetic approaches.

In terms of main early-branching euteleost lineages, only the placement of Argentiniformes between concatenated and GT-ST hypotheses varied. The placement of the argentiniform fishes is unresolved by this study and the branching of the Neotelostei, Stomiatii and Argentiniformes may be considered a soft polytomy. We find that phylogenomics and the application of the coalescent model in phylogenetics strengthen support for the earliest splits in the euteleostean radiation. However, key aspects of early euteleost phylogeny remain unresolved and leave open the question of whether extant genomes from these lineages retain historical signal that can be retrieved above the noise accumulated over hundreds of millions of years of independent evolution.

## ACKNOWLEDGEMENTS

We would like to acknowledge the following institutions and individuals that contributed to sampling in this project: Burke Museum of Natural History and Culture, Kansas University Biodiversity Institute and Natural History Museum, Academy of Natural Sciences of Philadelphia, Robert Marcotte (Univeristy of Alaska Fairbanks), Molly Hallock (Washington Department of Fish and Wildlife, retired), Motohiro Kikuchi (Chitose Salmon Park) and Peter Unmack (University of Canberra). Sébastien Lavoué (National Taiwan University), Thaddaeus Buser (Oregon State University) and Kerry Reid (University of California Santa Cruz) provided very helpful comments on drafts of the manuscript. We would like to thank Kevin McCracken (University of Miami) for his support in promoting Next-Generation Sequencing and Bioinformatics at the University of Alaska Fairbanks.

### Funding

This study was funded by a Genomics Seed Grant from the Alaska Idea Network for Biomedical Research (INBRE) awarded to MAC and JAL. Research reported in this publication was supported by an Institutional Development Award (IDeA) from the National Institute of General Medical Sciences of the National Institutes of Health under grant number P20GM103395 and by NSF DEB 0842397 to MEA. The content of this manuscript is solely the responsibility of the authors and does not necessarily reflect the official views of the NIH. The funders had no role in study design, data collection and analysis, decision to publish, or preparation of the manuscript.

### Grant Disclosures

The following grant information was disclosed by the authors:
Alaska Idea Network for Biomedical Research (INBRE).
National Institute of General Medical Sciences of the National Institutes of Health: P20GM103395.
NSF DEB: 0842397.

### Competing Interests

The authors declare there are no competing interests.

### Author Contributions

- Matthew A. Campbell conceived and designed the experiments, performed the experiments, analyzed the data, contributed reagents/materials/analysis tools, wrote the paper, prepared figures and/or tables, reviewed drafts of the paper.
- Michael E. Alfaro and J. Andrés López conceived and designed the experiments, contributed reagents/materials/analysis tools, wrote the paper, reviewed drafts of the paper.
- Max Belasco conceived and designed the experiments, performed the experiments, reviewed drafts of the paper.

### Data Availability

Assembled sequences are included in the Data S1. Data available from the Dryad Digital Repository: 10.5061/dryad.7p511.

### Supplemental Information

Supplemental information for this article can be found online at http://dx.doi.org/10.7717/peerj.3548#supplemental-information.

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
