# Peer review of "Early-branching euteleost relationships: areas of congruence between concatenation and coalescent model inferences"

_PeerJ, doi:10.7717/peerj.3548_

## Round 0.1 · original submission · Minor Revisions

In this manuscript the authors use 53 loci identified as ultra-conserved elements to infer relationships among early branching euteleostean fishes. A robust data set of 17,957 aligned characters for 34 taxa is analyzed using Maximum Likelihood of a concatenated data set, as well as gene-tree species tree analyses of individual genes. Happily, the results are similar for the two analyses, and support some previously proposed relationships, although there are 6 taxa spread among 3 clades in which placement differs between analyses. In contrast to many other studies with conflicting data, the differing placements between analyses are, in this case, robustly supported. Both external reviews were complimentary, and suggested the manuscript be accepted pending minor revisions, and I concur. The comments by the reviewers are useful and constructive, and should be addressed point-by-point in the cover letter of the revised manuscript. The one area where the reviewers, particularly reviewer 1, had concerns was the lack of data/analytical exploration to explain the robust conflicting support between analyses. What is contributing to this conflict? Is it a result of faulty assumptions implicit or explicit in one of the analytical methods, or lack of fit of the data to the assumptions of one or both of the chosen methodologies? There are clearly branch-length issues in the data set, and other factors such as differing levels of ASRV, base compositional shifts, missing data, missing loci, poorly aligned regions, particular badly-behaved genes could be explored. This is a great case in which to explore these issues, since the conflict is robustly supported. Of course, the authors may feel that this is their paper, and they get to decide what they will focus on, and I agree. So, I certainly don't insist on these analyses, but if they don't do them, someone else surely will. Why not make the most of their own data?

A couple of minor editing suggestions below

Line 186-188: Through the additional concatenated analyses presented in the Supplemental Document S1, conflicts between ML and GT-ST results presented in Figures 1 & 2 are shown to be the product of the distinct analytical frameworks and do not result from data modeling.

Line 298
The Protacanthopterygii is an historically important taxon

Reviewer 1 ·

Basic reporting

Basic Reporting:
1) Clear, unambiguous, professional English language used throughout: The writing and language are fine throughout.
2) Intro & background to show context. Literature well referenced & relevant: The intro and background are fine, and the literature generally is well-referenced.
3) Structure conforms to PeerJ standards, discipline norm, or improved for clarity: The manuscript appears to conform to PeerJ standards.
4) Figures are relevant, high quality, well labelled & described: Yes, the figures are fine. I would prefer, for clarity and ease of reading, that some of the supplementary online figures be put in the main text so that trees from various analyses can be easily compared without going to the online material.
5) Raw data supplied (see PeerJ policy): Yes, I was easily able to access the data that underlies the phylogenetic trees that were presented.

Experimental design

Experimental Design
1) Original primary research within Scope of the journal: Yes.
2) Research question well defined, relevant & meaningful. It is stated how the research fills an identified knowledge gap: Yes.
3) Rigorous investigation performed to a high technical & ethical standard: Yes, but see comments below on further suggested analyses to try to discern why concatenation and coalescence results do not agree.
4) Methods described with sufficient detail & information to replicate: For the most part, yes. One concern I had was whether branch length heterogeneity was accounted for in the BEAST analysis as it was in the *BEAST analysis. The authors note that any differences in the BEAST and *BEAST that are due to models of molecular evolution are accounted for by using the same partitioning scheme and substitution models in each of these analyses. This leaves ILS as the only difference. But, were branch lengths of genes modeled in the same way for each analysis? From my experience, much (most?) of the difference in likelihood scores between concatenation and coalescence is not due to accounting for different topologies for different gene trees (i.e., allowing for ILS or other processes like gene flow that create conflicts in topology among genes). Instead, the better likelihood of coalescence models are often driven strongly by the many more branch length parameters in the coalescence model (unique branch lengths for each gene tree) rather than a single common set of branch lengths for all gene trees in the concatenation tree. It would be best to run a Bayesian or ML concatenation analysis in which unique branch lengths were permitted for each gene tree (instead of one set of branch lengths for all genes or a rate multiplier for branch lengths among different genes), as is the case in many coalescence methods, to see if results converge.

Validity of the findings

Validity of the Findings:
1) Impact and novelty not assessed. Negative/inconclusive results accepted. Meaningful replication encouraged where rationale & benefit to literature is clearly stated: New data and analyses and conclusions were presented in a coherent way.
2) Data is (…are?) robust, statistically sound, & controlled: The data seem fine. Interpretation of the results could perhaps require a few more analyses and interpretations and scrutiny (see below). Generally no strongly opinionated statements were made regarding interpretation of the differences in topology among analyses, but there is a hint that the authors think ILS is to blame and that the *BEAST analysis is preferred.
3) Conclusions are well stated, linked to original research question & limited to supporting results: For the most part, things are fine, but I think that more detailed and inquisitive analysis of the data should be executed, and this could lead to better insights into the conflicts between coalescence and concatenation results (see below).
Speculation is welcome, but should be identified as such: There is some speculation at the end and sprinkled in earlier that the conflict between concatenation and coalescence analyses is due to ILS problems. But, little analysis or evidence is presented to argue for this over the coalescence analysis being the problem. *BEAST analysis is quite sensitive to rooting issues, because it is a clock-based method and violation of its very specific assumptions can lead to problems. For example, in the initial description of the method (Heled and Drummond, 2009), a single PCR contaminant in the gopher dataset that was analyzed, a mixup of an ingroup sequence with an outgroup sequence, led to an error in the inferred species tree which goes away when the contaminated locus is deleted from analysis. In the original description of the gopher data in Syst. Biol. (Belfiore et al., 2008), the authors predictably blamed concatenation for problems instead of errors in their own data that caused a problem for *BEAST analysis but not concatenation analysis. This type of thing is unfortunately quite common in high impact papers. I have looked at the data now in ten or so high impact papers where it was claimed that concatenation was getting the wrong result and coalescence got the right result and solved an intransigent problem in systematics. In each case, basic errors in data management (contamination, misalignment, etc.) or basic errors in analysis (poor methods employed, poor searches executed, etc.) were the cause of the problem, not concatenation. The current paper is quite understated and does not argue strongly for one result over another, but the paper would benefit, I think, from taking a more critical look at the basic underlying data which drives the differences among trees. For example, the authors note 3 critical differences between ML concatenation and *BEAST coalescence trees, and note that BEAST concatenation does not resolve the conflict at one of the three controversial splits. But, BEAST concatenation does agree with *BEAST coalescence at the two other conflicts and disagrees with ML concatenation. So, perhaps using a molecular clock, as in the various BEAST/*BEAST runs, is the problem, not concatenation. Indeed, the three spots where there are problems are areas with extreme rate heterogeneity in terms of ML branch lengths (see Figure 1), and misrooting by assuming a clock could be occurring at each of these nodes. A similar pattern emerged in a recent paper by Lambert et al. (2015) where BEAST and *BEAST agree (‘When do species-tree and concatenated estimates disagree? An empirical analysis with higher-level scincid lizard phylogeny’). In Figure 1 of Lambert et al. (RAxML tree), it is clear that there is a large rate speed up in the Lygosominae clade and that midpoint rooting the tree would give something more like the clock-constrained BEAST and *BEAST analyses. Perhaps the clock-like analyses are right, or perhaps both are just misrooting badly in the same way due to this crazy rate difference. At the three conflicts in the fish tree of interest, I would argue that a similar misrooting might be occurring at each of the three conflicts. The relationships supported by *BEAST always ‘rectify’ the branch length inequalities at the three conflicting spots. In running a simple equally weighted parsimony analysis of the fish dataset, the resulting tree exactly (I think) matches the ML concatenation result despite all of the various long branches and rate inequalities. By simply mapping synapomorphies to nodes, I found that many of the characters that support the concatenated ML and parsimony trees’ result of Acanthurus + Antennarius (long branched) are confined to just one gene (locus 1053). It would be profitable to rerun various analyses with single loci deleted to see if any particular locus drives the conflicting results. For example, some of the alignments are a bit dicey in spots and even small slips in alignment can drive results at tight internodes where there may be just a few informative characters (and a set of linked misleading characters due to just one alignment error). Unfortunately, phylogenomic analyses, at tight internodes, can be biased by such simple errors in data rather than anything interesting (such as accounting for ILS or not). So, I would recommend tracking exactly which characters and which genes optimize to the conflicting internodes in different trees (using parsimony or likelihood); are the supporting characters in areas of bogus alignment, or not. Also, it would be good to run gene trees of the 50 or so loci to see exactly which gene trees, if any, support the controversial nodes at three spots that conflict between concatenation and coalescence. I would argue that way too much attention generally is given to just staring at posterior probabilities or bootstrap scores at conflicting nodes, and much more attention should be spent looking at the underlying data and the assumptions of the underlying methods and how these two factors interact. It is clear that many of the most prominent ‘leaders’ in our field have argued recently that alignment or choice of methods don’t matter or even that a 50% bootstrap score is not different than a 95% bootstrap score (e.g., see Edwards et al., 2016), but of course such arguments should be mocked by any serious worker in the field. I think that just a little bit more analysis in this case study could lead to more understanding of the conflict and perhaps even derive a convincing conclusion on the matter. So, I would suggest:
1) seeing which characters and which genes and which gene trees agree/disagree at the 3 critical conflicts noted by the authors.
2) execute subsampling of the data by removing one or more loci from analysis and rerunning – might determine which genes drive the results. This might not be worth the effort, but I suspect at least one of the conflicting nodes will go away with this approach of removing a single gene.
3) delete any sketchy alignment regions from analysis and rerun the RAxML and BEAST concatenation analyses to see if anything changes toward congruence, or not, with *BEAST. Do same for coalescence analysis.
4) run some summary coalescence methods (ASTRAL, MP-EST, STAR, etc.) on gene trees and see if get same conflicting nodes that you get for *BEAST. Is the conflict seen due to concatenation/coalescence differences or is just *BEAST coalescence and BEAST concatenation which assume a clock getting these answers. Just eyeballing the branch lengths in Fig. 1, rate heterogeneity looks like a possible problem, as is often the case at high divergences among sequences.

Additional comments

Nothing beyond what is stated above.

Reviewer 2 ·

Basic reporting

This was one of the most well-written and clear papers I have reviewed in quite some time. The scholarship includes citations to early morphological work as well as recent molecular work and frames the problem of resolving euteleost relationships very well. The figures are clear and easy to read. Likewise, results are organized well and the flow of the discussion is logical.

Experimental design

This manuscript suffers from no obvious phylogenetic pitfalls. Caveats and model assumptions are plainly discussed. The authors comparison of gene-tree species-tree methods against concatenation is appropriate. While there are certainly other approaches that could extend this work further (phylogenetic information content of loci, model performance, etc) these are not necessary in this work.

The work here creates a roadmap for future work that is needed for resolving deep branches in the ray-finned fish tree of life, simultaneously highlighting the consistent placement of lineages like Lepidogalaxias and areas of uncertainty.

All methods are described in detail and this study could be easily replicated from the given detail.

Validity of the findings

The findings rest on strong UCE-based data and standard phylogenomic inference methods. The authors are very careful to not overstate any findings.

Additional comments

I really enjoyed this paper and have only two suggested revisions for the discussion.

1) At the end of line 278, would the authors be able to extend the discussion on conflict to include insights from the new Shen et al. paper in Nature Ecol. Evol (Contentious relationships in phylogenomic studies can be driven by a handful of genes). This seems an alternative explanation for the strongly supported conflict in the data.

2) Line 338. I do not think that the wording here is right. The final question to consider is not a matter of preference in my opinion, this seems misleading. I understand the meaning the authors intended and agree with the rest of the paragraph but think that this opening sentence should be cut. The real resonating question is the last sentence of the paragraph/manuscript and this opening is pretty distracting.

Minor:

line 29: Lepidogalaxias spelling

Figure legends: Break the long sentence on rooting into two: "The tree is rooted on Polypterus senegalus. This taxon, Amia...."

---

## Round 0.2 · accepted · Accept

In this revision, the authors have carried out additional analyses that shed light on at least some of the incongruence among methods, added a new figure that nicely summarizes areas of disagreement, cited some recent germane papers, and revised the discussion based on these new results. In my opinion, the manuscript should now be accepted for publication. I append a few editorial suggestions bellow:

line 272:resolves the otomorphs as found
line 360: summary coalescent analyses
line 453: The placement of the argentiniform fishes is unresolved by this study and the branching of the Neotelostei, Stomiati and Argentiniformes may be considered a soft polytomy